# Multi-Parameter Optical Monitoring Solution Applied to Underground Medium-Voltage Electric Power Distribution Networks

**DOI:** 10.3390/s23115066

**Published:** 2023-05-25

**Authors:** Fabio R. Bassan, Joao B. Rosolem, Claudio Floridia, Rivael S. Penze, Bruno N. Aires, Ronaldo A. Roncolatto, Rodrigo Peres, João R. Nogueira Júnior, João Paulo V. Fracarolli, Eduardo F. da Costa, Filipe H. Cardoso, Fernando R. Pereira, Carla C. Furoni, Cláudia M. Coimbra, Victor B. Riboldi, Camila Omae, Marcelo de Moraes

**Affiliations:** 1CPQD—Research and Development Center in Telecommunications, Campinas 13086-902, SP, Brazil; rosolem@cpqd.com.br (J.B.R.); floridia@cpqd.com.br (C.F.); rpenze@cpqd.com.br (R.S.P.);; 2CPFL Energia, Campinas 13088-900, SP, Brazil

**Keywords:** medium voltage, distribution network, underground power cables, monitoring system, FBG, optical fiber

## Abstract

This work presents a multi-parameter optical fiber monitoring solution applied to an underground power distribution network. The monitoring system demonstrated herein uses Fiber Bragg Grating (FBG) sensors to measure multiple parameters, such as the distributed temperature of the power cable, external temperature and current of the transformers, liquid level, and intrusion in the underground manholes. To monitor partial discharges of cable connections, we used sensors that detect radio frequency signals. The system was characterized in the laboratory and tested in underground distribution networks. We present here the technical details of the laboratory characterization, system installation, and the results of 6 months of network monitoring. The data obtained for temperature sensors in the field tests show a thermal behavior depending on the day/night cycle and the season. The temperature levels measured on the conductors indicated that in high-temperature periods, the maximum current specified for the conductor must be reduced, according to the applied Brazilian standards. The other sensors detected other important events in the distribution network. All the sensors demonstrated their functionality and robustness in the distribution network, and the monitored data will allow the electric power system to have a safe operation, with optimized capacity and operating within tolerated electrical and thermal limits.

## 1. Introduction

Underground distribution systems are reliable ways of distributing electricity, even though they have a much higher installation cost than aerial cables. The most important element of the underground distribution system is the conductor cable, which must operate for more than thirty years in a reliable way [1,2].

If, on the one hand, the temperature supported by the insulating material and the cable itself are factors that limit the load current and power, on the other hand, the conductors of underground power systems are exposed to electrical, mechanical, and thermal stresses during operation, which can reduce the service life or even damage the cable insulation [3,4]. Damage to the cable can result in the degradation of the insulation, the occurrence of partial discharges, and, finally, the rupture of the insulation.

Transformers housed in underground chambers or pad-mounted enclosures (on the ground), splices, connectors, and junction boxes are other critical elements in the underground power distribution system [5,6].

The use of monitoring in the underground distribution systems allows the electric power system to operate safely, with optimized capacity, and operate within tolerated electrical, thermal, and mechanical limits. In doing so, utility companies can reduce the frequency of preventive maintenance in their underground networks, increasing availability and reducing operating costs. It is necessary to implement monitoring of critical components of the underground distribution system, such as cable, cable connections, underground boxes, and transformers. Utility companies can reduce the frequency of preventive maintenance and mitigate the need for corrective maintenance in their underground networks by adopting real-time monitoring to improve the assessment of the health of the underground distribution network. The monitoring system increases availability, reducing operating costs and avoiding the need for interventions as much as possible in the road systems of urban areas. 

Some works in the literature proposed different types of monitoring systems for underground distribution power systems. In [7], the condition monitoring and diagnostic techniques were revised for underground power cables. In [8], the development and improvement of an intelligent cable monitoring system for the underground distribution power system were proposed using DTS (distributed temperature sensing) technology and OFCPC (optical fiber composite power cables) as the underground distribution lines in the field. In [9], new developments in sensors, IED (intelligent electronic devices), and communication technologies were proposed as attractive alternatives for monitoring the underground distribution network in general and in cable chambers/manholes in particular. In [10], the second generation of an underground sensing and monitoring system was introduced and discussed, and the results and the conclusions of the tests performed by IREQ (Hydro-Quebec, Varennes, QC, Canada) were presented. In [11], a distributed sensor network was proposed, based on several nodes deployed over underground distribution lines useful in urban environments. In [12], a monitoring system was proposed, capable of detecting and classifying the health conditions of MV (Medium-voltage) underground cables.

This work presents an optical fiber multi-parameter monitoring solution applied to an underground power distribution network, in which a major part of the network elements is monitored passively using integrated optical fiber technology. The technology integrated with optical sensors unifies the network, the sensing system, installation procedures, calibration, and operating procedures. This proposed system can bring a reduction in operation costs and increase safety.

The monitoring system demonstrated herein uses Fiber Bragg Grating (FBG) [13] sensors to measure multiple parameters, such as the distributed temperature of the power cable, the external temperature of the transformers, the current transformers, liquid level, and intrusion in the underground junction boxes. Partial discharges (PD) of cable connections were monitored using sensors that detect radio frequency PD signals followed by a laser diode modulation (PD-LD). All these monitored parameter signals were integrated into a single optical fiber wavelength division multiplexing (WDM) sensing network.

There are several advantages intrinsic to optical fibers [14,15], such as high sensitivity, immunity to electromagnetic interference, wide bandwidth, resistance to harsh environments, ease of installation, and long lifespan, and for FBG sensors, the network-multiplexing capability of different sensors in a single fiber is also an advantage of optical sensors. In addition, electrical power from an alternated current (AC), batteries, or solar panels is not necessary. In terms of detection time, the optical fiber sensor’s advantage arises from the network’s reliability, as the sensor information is inside the optical fiber, and the distance or line-of-sight between the sensor and the interrogator is not a problem like it is for wireless sensors, for example. Network availability is also an advantage for optical sensors in terms of detection time. Concerning accuracy, FBG sensors can detect even small changes in physical parameters, and in addition, the information measured by the sensor is in the wavelength variation (frequency) and not in the amplitude, which maintains the fidelity of the measurement without losses along the optical fiber due to signal attenuation.

Some works proposed FBG in underground distribution systems. In [16,17,18,19], FBG temperature sensing was proposed for underground cables. In [16], the authors demonstrated this technique in the laboratory, using discrete sensors; in [17], it was demonstrated in the field using an FBG array sensor; in [18], a new packaged FBG sensor was developed for underground cable temperature monitoring; and in [19], FBG sensors were used to monitor temperature power cable joints in underground utility tunnels. In [20,21,22], the authors studied the FBG as a PD sensor for cable applications. In [23], the authors developed an optical voltage transformer based on FBG-PZT for power quality measurements, and in [24], FBG sensors were studied for enhanced microgrid performance.

Because the FBG sensors are still not yet too sensitive to detect PD in underground cables [21,22], we used in this work the technique PD-LD, as this technique is very effective to monitor partial discharges [25,26]. In the monitoring system described herein, this technique adds energy harvesting based on a current transformer to supply the laser circuit.

In the following, we describe the optical multi-parameter monitoring system applied to a medium-voltage underground distribution network. The main contribution of this paper, to the best of our knowledge, is that it is the first demonstration of an optical multi-parameter system applied to a medium-voltage distribution network using FBG sensors with laboratory and field results. We also highlight the method that permits a current sensor with temperature compensation using Terfenol-D with a single FBG, and the optical network topology deployed that allows a partial discharge sensor based on energy harvesting and laser as light emission share the same optical fiber of sensors based on light reflection (FBG). In Section 2, the topology of the monitoring system is described, as well as the sensors and the monitored distribution network. In Section 3, we describe the characterization of the sensor in the laboratory and the results of the monitored networks for six months in two locations in the city of Campinas, Brazil. In Section 4, we discuss the results obtained in the laboratory and the field tests.

## 2. Materials and Methods

In this section, we present the sensors used in the system and the details of the monitored medium-voltage distribution network.

### 2.1. The Proposed Monitoring System

Figure 1 shows the generic idea of the proposed monitoring system. The interrogator is placed in a substation control room some kilometers away from the underground distribution lines. Optical aerial cables belonging to the utility company are used to transport the interrogator signals to monitor underground distribution lines and elements. 

### 2.2. The Interrogator

Figure 2a shows the block diagram of the interrogator of the monitoring system. As previously mentioned, the monitoring system is based on FBG and PD-LD technologies. In this way, the interrogator is composed of two different approaches. For FBG sensors, we used a commercial module (Hyperion si155 supplied by LunaInc, Roanoke, VA, USA) based on tunable laser technology. This module has four channels, a sweep rate of 1 kHz, and works in the spectral range from 1500 to 1600 nm. The PD-LD module is composed of a 155 MHz optical receiver followed by an RF detector based on an AD8310 integrated circuit and a DAQ board (DAQ-USB-6361 supplied by National Instruments, Austin, TX, USA). Both modules send the data to the main computer, where the data are analyzed and recorded by a self-developed LabVIEW application. Figure 2b shows the modules of the interrogator.

### 2.3. The Sensors

The sensors used in the monitoring system were developed to measure pad-mounted transformer currents and temperature, the distributed temperature of the medium-voltage cable, liquid level and intrusion in the manhole, and partial discharges in the medium-voltage connectors. The optical grid of the FBG sensor is from 1500 to 1600 nm.

The FBG array, temperature, liquid level, and intrusion sensors are commercial parts; the current sensor is homemade but uses a raw FBG. All FBG sensors and raw FBGs were supplied by Atgrating, Shenzhen, China; they also supplied the FBG sensor datasheet with all the FBG sensors’ characteristics. 

The FBG array sensor (Figure 3a) was used for the ampacity evaluation of the medium-voltage conductors. The sensor array was designed to use optical fiber coated with glass fiber-reinforced polymer (GFRP) to withstand the installation process. To increase the mechanical strength of the FBG array, it was installed in a micro duct. The sensor array used has 2 m FBG spacing.

The FBG temperature sensor was used to measure the external temperature of the pad-mounted transformer. Figure 3b shows the temperature sensor packing.

The FBG liquid level (Figure 3c) was designed to measure up to 2 m liquid height. In addition, the sensor has an FBG temperature sensor used to compensate for the liquid level measurement for the environmental temperature effects. In addition, the FBG temperature sensor allows the evaluation of the temperature in the manhole.

The intrusion sensor is based on the FBG displacement sensor fixed on the manhole entrance to detect the manhole cover displacement. Both liquid level and displacement FBG sensors have been demonstrated in previous field installations [27]. Figure 3d shows the displacement of the FBG sensor packaging.

The current sensor is used to measure the current on the secondary windings (low voltage) of pad-mounted transformers. The sensor package (Figure 3e) is based on a single FBG attached to a Terfenol-D bar as a sensing element [28]. This set is placed in a specially designed optical tray, which in turn is coupled to a package containing two pieces of nanocrystalline ferromagnetic material to concentrate the magnetic field from the electrical current through the conductor. A gap between the two pieces of the nanocrystalline material enables the magnetic field to expand and excite the magnetostrictive bar of Terfenol-D. This setting allows installing the proposed sensor in existing conductors without interrupting the energy deployment. In other words, this design allows using the proposed current sensor in the same way as a conventional current clamp meter. Another advantage is the separation of the magnetic field concentrator packaging (the nanocrystalline holder) from the optical tray where the sensing element is present. Compared with [28], in this new configuration of the sensor, an applied bias was introduced by a permanent magnet, which enables reading the waveform without frequency doubling. 

The PD-LD sensor (Figure 3f) was previously demonstrated in [16]. The sensor unit consists of an antenna connected directly to a semiconductor laser diode (LD). In previous works, a power-over-fiber technique was used to properly provide the bias polarization of the laser to have more PD sensitivity. In the current work, we used a current transformer clamp to the medium-voltage cable to provide laser diode bias. The antenna we chose for the sensor obeyed the following requirements: reduced dimensions adjusted for installation next to the medium-voltage connectors, resonance frequency within the spectrum band of the PD measurements (100 MHz), and the largest bandwidth possible. The impedance was adjusted for laser impedance matching. The sensor also has good directivity and it is electrically isolated. The antenna has the following characteristics: dielectric type = FR4 (print circuit), dielectric thickness = 1.6 mm, line width = 1 mm, lines separation = 1 mm, and the total antenna length = 3916 mm. The sensor was packaged with epoxy resin, which provides good mechanical and thermal resistance and electrical isolation. The connection of PD-LD sensors in the same fiber of the FBG sensors each laser of PD-LD sensor operates in a specific coarse wavelength division multiplexing (CWDM) wavelength. Each PD-LD sensor is connected to the fiber using an add-drop multiplexer.

### 2.4. The Monitored Underground Distribution Network

The monitoring system was installed in two underground distribution networks. Both networks are placed some kilometers from the control room. We used the optical fiber telecom networks belonging to the utility company to transport the signal from the monitored place to the control room.

#### 2.4.1. Downtown Underground Distribution Network Monitoring

Figure 4a shows the single-line diagram of the monitoring system installed at the downtown underground network and the images of the sensors installed in the distribution network. The interrogator was installed in an electrical substation about 2 km from the sensor installation site. The link that connects the interrogator to the sensors network uses aerial optical cables from the electric utility company. One hundred temperature sensors were installed, divided into two FBG array sensors of 100 m each, monitoring 200 m of underground distribution network. An FBG array is responsible for monitoring the path between CI-2/15 and CD-12, passing through CD-14 and CD-13. The other FBG array monitors the path from CI-2/15 to the TR1 transformer. In addition to the duct temperature sensors, liquid level, intrusion, and partial discharge sensors were also installed on CI-2/15 and current sensors on the TR1 pedestal transformer. 

Figure 4b shows the displacement sensor installed below the manhole door, while Figure 3c shows the liquid level sensor installed inside a tube in the manhole wall. Figure 3d is a picture of the PD-LD sensor installed near a medium-voltage connector, and finally, in Figure 3e, it is possible to see the temperature sensor fixed on the pad-mounted transformer wall, as well as the current sensors plugged into the low-voltage conductors of phases A, B, and C.

In this installation, an FBG interrogator based on laser sweeping with four simultaneous channels and a 1 kHz sampling rate, in addition to the signal reception system from the partial discharge sensors, was used to acquire and process the optical signals from sensors. The distribution of sensors per channel, as well as the wavelength of each sensor, is shown in Table 1.

Due to the limitation of the number of optical fibers and considering the spectral window of the FBG interrogator (1500–1600 nm) in this installation, it was necessary to use additional optical components in the optical network. Add-drops were used to allow the partial discharge sensors (operating in 1470 nm, 1490 nm, and 1610 nm) to share the same optical fiber as the intrusion and liquid level sensors (channel 3 of Table 1). In other words, add-drops were used for the multiplexing of the PD optical signals. Figure 5 shows the channel 3 optical configuration.

#### 2.4.2. Condominium Underground Distribution Network Monitoring

The single-line diagram of the monitoring system is shown in Figure 6a and the sensor photos in each installation place are shown in Figure 6b–d. In Figure 6b, we show the intrusion sensor; in Figure 6c, the liquid level sensor; and in Figure 6d, the temperature sensors, the PD-LD, and the current sensors. The interrogator was installed in a control room 500 m from the sensor installation place. As in the previous installation, one hundred temperature sensors were installed, divided into two FBG array sensors of 100 m each, monitoring 200 m of underground distribution. One of the FBG arrays is responsible for monitoring the temperature of the path between the TR1 point and the CP1 point, passing through the CP2 point. The other FBG array monitors the temperature from the TR1 to the TR2 points, passing through the CP3 point. The liquid level and intrusion sensors were installed in the CP2 point, and partial discharge, current, and temperature sensors were installed on the TR1 pad-mounted transformer.

In this installation, another FBG interrogator based on laser sweeping with four simultaneous channels and a 1 kHz sampling rate, in addition to the signal reception system from the partial discharge sensors, was used to acquire and process the optical signals from sensors. The distribution of sensors per channel and the wavelength of each sensor are shown in Table 2.

Due to the availability of optical fibers, in this installation, it was possible to use dedicated fibers that transmit the signals from the DP sensors directly to the receiver, so there was no need to use add-drops.

## 3. Results

### 3.1. Sensors’ Characterization in Laboratory

#### 3.1.1. Homemade FBG Current Sensor

The characterization of the FBG current sensor follows the procedure previously reported in [28]. In this configuration of the sensor, however, the applied bias by a permanent magnet enables the recovery of the waveform without frequency doubling, as can be seen in Figure 7 and Figure 8. Three optical sensors named S1, S2, and S3 were fabricated and characterized to be applied in each phase of the conductor. Figure 7 shows the response of the current sensor S1 in terms of wavelength for a fixed temperature of 20 °C and varying currents (values of 26, 130, and 312 A, in the plot). In Figure 8, the current was fixed (illustrated for 130 A), and the temperature was varied to 20, 40, and 60 °C.

In the previous work [28], it was demonstrated that with the values of Δλ and the minimum wavelength (λmin), is it possible to obtain the local temperature and current. In fact, from Figure 7 and Figure 8, we can observe that both Δλ and λmin vary as a function of temperature and applied current. Thus, a complete characterization of these parameters must be performed to recover from a wavelength waveform, the applied current, and the temperature.

The characterization was performed by recovering the wavelength waveforms and obtaining the Δλ and λmin for temperatures varying from 10 °C to 60 °C, in steps of 10 °C, and varying the current from 0 to 312 A in steps of 26 A. The results for sensor S1 are shown in Figure 9 for Δλ and in Figure 10 for λmin. 

All three sensors S1, S2, and S3 were tested by applying different currents and temperatures using the inverse procedure described in [28]. The currents were set to 25, 100, and 250 A and temperatures to 20, 30, and 40 °C. After applying the mentioned currents and temperatures, the values of Δλ and λmin were used to calculate the current and temperature. In Figure 11, we show the recovered current values for different applied temperatures. All three sensors responded properly in the tested range and slight values of currents were recovered. We observed that one of the causes of this variation is the positioning of the cable inside the sensor, as there is a gap in the magnetic core. The sensor design was made for a large cable compared with the one used in the laboratory.

In Figure 12, we plot the recovered temperature for different applied currents. All three sensors show good behavior, with 92% of the recovered values lying within ±1 °C. 

#### 3.1.2. Commercial FBG Array, Temperature, Liquid Level, and Intrusion Sensors

The characterization of the commercial FBG array was obtained by using an FBG interrogator. The FBG array spectrum with 50 peak wavelengths is shown in Figure 13. As can be seen, the 50 FBGs inscribed in the FBG fiber array are almost equally spaced by 2 nm, starting from 1500.5 nm to 1598.5 nm. The spectrum is well balanced with an almost flat response with a maximum deviation of 2.7 dB in the entire range.

The characterization proceeded with the measurements of the wavelength shifting with the applied temperature inside a climatic chamber. The temperature was varied from 0 to 70 °C. Inside the climatic chamber, two FBG array cables were tested, one with GFRP and the other inside a conventional optical fiber cable. A reference FBG was also used as a comparison. Figure 14 shows the results of the wavelength shift as a function of elapsed time for these different FBG array cables and reference FBG. As can be seen, the reference FBG and the FBG array in a conventional optical fiber cable have close wavelength shift results, with the FBG array having a slower heating process. On the other hand, the FBG array with GFRP resulted in higher wavelength shifting—in other words, the GFRP FBG array has greater temperature sensitivity.

This fact can be better understood by referencing Figure 15. In this figure, the wavelength shifting is displayed as a function of applied temperature after stabilization time. As can be seen, the reference and optical fiber cable have almost the same behavior, while the GFRP has a higher angular coefficient (sensitivity). The sensitives are 0.0098, 0.010, and 0.015 nm/°C for the reference FBG, optical fiber cable, and GFRP FBG array cable, respectively.

The punctual temperature sensor was characterized using a thermal cycle of heating and natural cooling. The wavelength shifting was compared with a reference temperature sensor (thermocouple), and the result is shown in Figure 16.

Another characterized sensor was the liquid level sensor, in which the wavelength shifts as a function of time when increasing the water level from 0 to 1.5 m, as is shown in Figure 17. The wavelength shifting is negative and reaches −0.36 nm for a water column level of 1.5 m. 

Finally, the intrusion sensor was tested by performing three displacement compressions of 50 mm, resulting in a wavelength shift of 1.4 nm, as shown in Figure 18. This wavelength shifting is more than sufficient to detect a state of intrusion during an opening of the manhole. This sensor will work through threshold detection. In other words, if the initial wavelength shifts more than a predefined value, an intrusion is reported.

#### 3.1.3. Homemade PD-LD Sensor

The homemade PD-LD sensor was characterized in terms of optical power as a function of conductor current and in terms of average PD level as a function of the distance of the PD source. As shown in Figure 19a, the sensor will be operational if the conductor current is above 50 A, resulting in a constant optical power emission of −5 dBm. The sensor operates up to 50 cm from the PD source (in the air, with no obstacles), as shown in Figure 19b. Therefore, the distance from the suspicious PD source to the sensor must be taken into account where the sensor is placed.

### 3.2. Field Evaluation

#### 3.2.1. Acquisition and Processing and Visualization Software

The data acquisition and visualization tools were developed in LabVIEW (2017) and used in the field tests. Some software dashboards are shown in Figure 20 and Figure 21. Since LabVIEW works by parallelizing data flow, the software is made up of several loops running in parallel, each with a specific function, as follows:Reading buttons and controls to monitor user inputs;NI-DAQ (Data Acquisition) device periodically reads the high-speed interrogator data;State machine controls software actions according to the state read from the state queue (initialize, configure, measure, and error);Periodically read the FBG interrogator output used for the low-speed sensors;Data treatment and display that applies the necessary mathematical operations to the measured signals and displays the results on a dashboard;DNP3 communication with the DNP3 server;Local storage of measurements for software analysis (possible debugs);Send commands to the high-speed interrogator firmware. In normal operation, the firmware is already calibrated. This loop is used if a local configuration is required.

#### 3.2.2. Data Analyses from the Field Tests

The whole system was deployed in July 2022 in the downtown area and at the condominium site. Data have been collected since then. The results of the various parameters are shown and discussed in the following section.

##### Downtown Underground Distribution Network

The measurements of currents and temperatures in the three secondary phases of the transformer in the downtown underground distribution network of the Francisco Glicerio Avenue installation are shown in Figure 22.

The FBG arrays 1 and 2 were installed on Francisco Glicerio Avenue in the downtown underground network and monitored 200 m of the underground network. Table 3 shows some highlights of the FBG arrays.

Figure 23 shows the evolution of the temperature measurements of the FBG Arrays 1 and 2. We highlight that at the beginning of spring in Brazil (09/23), there is an increase in the temperature measured by the FBG array. 

In Figure 23, corresponding to FBG Array 1, it is possible to notice the periodic variation in the temperature measured close to 55 and 95 m. These points correspond to underground manholes CD-14 (55 m) and CD-13 and CD-14 (95 m). At these points, the FBG array is more exposed to environmental temperature variation. For this reason, the periodic temperature variation is more evident in these areas. During the evaluated period, an increase in temperature occurred at the end of October, with a sudden and short drop at the beginning of November.

Figure 23 also shows the evolution of the temperature measurements of FBG Array 2. As in the previous case, an increase in the temperature is observed at the end of October after the beginning of the spring. It is also possible to notice in Figure 23 the periodic variation in the temperature measured close to 60 and 98 m. These points correspond to the pad-mounted transformer and the manhole S2-31. 

According to Brazilian standards [29], the maximum current supported by the conductor is a function of the internal duct temperature, and above 20 °C, the maximum current supported by the conductor must be reduced to avoid conductor damage (this standard is based on [30,31]). Therefore, temperature monitoring in an underground network optimizes the use of the legacy network and postpones investment in new underground networks.

Figure 24 shows the temperature measurements taken on the surface of the TPU300 transformer during the analyzed period. The transformer temperature variations are mainly linked to the variation in the environmental temperature, shown in the fast variations in the measurements in Figure 24. 

The increase in temperature in late October also originates from the rise in environmental temperature due to the beginning of spring. A great temperature drop at the beginning of November was also a climactic effect, as can be seen in the meteorological data shown for comparison. These rise/drop events of late October and the beginning of November can also be seen in the temperature of the current sensor and from the measurements made by FBG Arrays 1 and 2.

Figure 25 shows the temporal evolution of the intrusion sensor measurements (wavelengths captured from August 2022 to January 2023). We highlighted in the graph the event T1, where there was an intrusion test with the opening of the manhole cover. It is observed that the wavelength exceeded the threshold established for intrusion detection.

The continuous amplitude variations in the measured signal are due to the environmental temperature variation. This sensor was installed below the cover of the manhole CS-2/15, a more sensitive area for the environmental temperature variation. Even with these variations, the sensor did not present false failures during the evaluated period.

Figure 26 shows the measurements of the liquid level sensor (in meters) and its temperature from August 2022 to January 2023. We highlight in the graph a calibration performed in the temperature measurement (C1). The temperature adjustment was made using a calibration instrument. The difference found was +6.6 °C. The compensation was made by adding +6.6 °C to the system measurement.

The liquid level exceeds the 0.10 m level at one time during the entire period of monitoring. This event is associated with intense rains in January 2023.

The partial discharge sensor works from a polarized laser connected to an element that captures electrical discharges. These discharges modulate the laser, causing this element to emit optical power. The emitted optical power is launched into the fiber and analyzed in the interrogator, generating discharge evolution graphs over time. A reference optical power level is expected from the interrogator, which indicates that the laser is polarized. Partial discharges, in turn, are visualized as optical power peaks that exceed a certain threshold previously recorded in the system.

During one of the visits to the place where the sensors were installed, the optical power received by the interrogator was initially measured. In this evaluation, the absence of optical power in all sensors was verified, indicating that the sensor’s laser was not operating above the threshold. The laser operating above the threshold is obtained through the energy-harvesting circuit that is collected from the medium-voltage conductors.

The currents of the conductors where the parts were installed were 15 A, 14.2 A, and 15.2 A for phases A, B, and C, respectively. The current in the conductors has a lower value than those used as a reference for the development of the sensor (around 50 A). Due to the low current in the conductor, the laser in the sensor does not work above the threshold. This is the reason for the lack of optical power in the interrogator. In this case, the sensor will capture only high-level discharges.

A second test was performed to verify the detection sensitivity of partial discharges by the sensors. In this case, the test was performed using a discharge generator close to the sensors. A third test was to use an RF detector near the medium-voltage connectors. In both tests, the sensors were not able to capture the phenomenon. A Appendix A is available online regarding this third test.

In conclusion, as is shown in Figure 27, during the evaluation period, no partial discharge events were observed in any of the three monitored phases.

##### Condominium Underground Distribution Network

Figure 28 shows the current and temperature measurements in the three secondary phases of the transformer at the condominium installation. The lack of data in the graphs corresponds to moments when the system was offline and data were not recorded. The maximum current occurs for phase B and corresponds to 220 A.

In Figure 29, we compare the current and temperature acquired at the condominium and on Glicerio Avenue. As can be seen, the current in both places is not in phase. This behavior is explained by the difference in load activity at these sites. The higher consumption occurs at night at the condominium when people return from work activities to their homes and use showers and air conditioners. Conversely, Glicerio Avenue is a commerce place where higher consumption occurs during the day. Regarding the temperature, we observe that both installation places mimic the data obtained from a meteorological station in the metropolitan area of the city.

Two factors explain the temperature behavior, sun exposition, and load characteristic of the transformers. If on the one hand, the transformer on Glicerio Avenue is placed below some trees and did not receive sun radiation during the day, then the load characteristic of the transformer is high during the day and low during the night due to this place being a commercial center, as there is more electricity consumption during the day. On the other hand, a transformer at the condominium receives direct sun radiation, and during the night, there is an increase in the load due to the higher consumption at night when people return from work activities to their homes and use high-consumption appliances. The sun radiation is responsible for the temperature offset during the day and the load of the transformer is responsible for the temperature offset at night.

Data on the semi-distributed temperature in the underground network installed at the condominium are given in Table 4.

Figure 30 shows the evolution of the temperature measurements of the condominium’s FBG Array 1, and FBG Array 2’s temporal evolution. It is possible to verify a direct relationship between the increase in environmental temperature and the temperature measured in the ducts. During the evaluated period, no temperature increase or decrease events were detected without a direct correlation to the environmental temperature variation. It is noticeable the day/night temperature variation for FBG Array 2 in the upper portion of Figure 30 corresponds to FBG46 to FBG50 placed outside the duct under Transformer 2.

According to Brazilian standards [29], the maximum current supported by the conductor is a function of the internal duct temperature, and above 20 °C, the maximum current supported by the conductor must be reduced to avoid conductor damage. Therefore, temperature monitoring in an underground network optimizes the use of the legacy network and postpones investment in new underground networks.

Figure 31 shows the temperature measurements performed on the surface of Transformer 1 during the analysis period. The transformer temperature variations are mainly linked to the environmental temperature variation for seasonal and daily periods, as can be shown in Figure 31 (for example, the increase in temperature after the beginning of the spring, as indicated by a dotted green line). 

Figure 32 shows the temporal evolution of the intrusion sensor measurements (wavelengths captured from August 2022 to January 2023). We highlighted in the graph the events T1 and T2, where there were two intrusion tests with the opening of the manhole cover, above where the sensor was installed. It is observed that the wavelength exceeded the threshold established for intrusion detection.

The continuous amplitude variations in the measured signal are due to the environmental temperature variation. This sensor was installed below the cover of the manhole CP2, where the environmental temperature variation is more evident. Even with these variations, the sensor did not present false failures during the evaluated period.

Figure 33 shows the measurements of the liquid level sensor (in meters) and its temperature from August 2022 to January 2023. During almost all this time, the liquid level in the manhole remains under 0.10 m, meaning that the manhole was empty. Regarding the measured temperature, during all this analyzed time, the temperature was close to 20 °C. We highlighted in the graph of Figure 33 a correction for the temperature measurement (C1). The temperature adjustment was made using a calibration instrument. The temperature was measured in the sensor location through a thermocouple. The measured temperature was compared to the system measurement. The difference found was +1.7 °C. Compensation was made by adding +1.7 °C to the system measurement. The liquid level exceeded the 0.10 m level two times during the entire period of monitoring. These events are associated with intense rains in January 2023.

Partial discharge measurements had similar tests as the downtown site. During one of the visits to the condominium, the optical power received by the interrogator was initially measured. In this evaluation, the absence of optical power in all sensors was also verified, indicating that the sensor’s laser was not operating above the threshold. The currents of the conductors where the partial sensors were installed were 1.3 A, 1.4 A, and 2.0 A for phases A, B, and C, respectively. However, the levels of discharge in the medium-voltage connectors used to connect to the transformers were high enough to be detected by the sensors, as shown in Figure 34. The discharges are more intense in phases A and B in terms of events and amplitude. 

A second test was performed to verify the detection sensitivity of partial discharges by the sensors. In this case, the test was performed using a discharge generator close to the sensors. A third test used an RF detector near the medium-voltage connectors. In both tests, the sensors were able to capture the phenomenon. A Appendix A is available online regarding this third test.

In September 2022, the detection level threshold was increased to reduce the intrinsic interferences. The data have been considered in the analysis from this month. It can be observed that some periods of higher levels of discharges were detected mainly in phase B during January 2023. 

## 4. Discussion

For the medium-voltage cable, according to [29], the maximum temperature in steady state is 90 °C; in case of overload, it is 130 °C and 250 °C in a short circuit for conductors with XLPE insulation. During the measurements, the maximum temperature recorded was 27 °C on Glicerio Avenue and 30 °C at the condominium, and, as previously shown, is directly related to the variation in the ambient temperature. No anomalies were detected during the period. Furthermore, according to [29], a correction must be applied to the value of the maximum current of the conductors according to the increase in temperature in the environment where the conductor is installed. This analysis is applicable through the value of the environmental temperature measured by the FBG array sensors, in such a way that it is possible to know the maximum current that can be applied to the cable, optimizing the network operation. In this way, temperature sensing along the conductive cable can indicate punctual faults in the cable or network elements and is useful for verifying the ampacity of the circuit being monitored. A punctual fault in the cable could affect the conductivity of the conductor material (copper or aluminum) at this point. The conductive change will increase the temperature and can degrade the cable insulation, causing current leakage and even a short circuit between the conductor and armoring of the medium-voltage cable. Monitoring the temperature along the cable can prevent current leakage and even a short circuit, avoiding electrical network unavailability, and allows optimizing the cable ampacity, as the current carrying capacity in the electrical cable is linked to the ambient temperature.

When we analyze the information on the evolution of partial discharges, they give us, for example, information about the health of the monitored element, since partial discharges are the first symptom that some anomaly is occurring in the material and, consequently, its identification allows predicting possible failures and acting in predictive maintenance. When we look at the information on partial discharges, it is important to mention that, according to several manufacturers, the useful life of medium-voltage elements under normal operating conditions is long, depending on the processes and constructive characteristics that maintain quality to satisfy the relevant standards. Through online measurements of partial discharges, it is possible to predict future actions for the cables and thus prepare a maintenance plan based on their condition. The way to evaluate partial discharges is to monitor their evolution over time; a rapid evolution in the volume of measured partial discharges is an indication of a possible failure. During the evaluation period, there were detected events that could indicate failures in the monitored elements (at the condominium in this case), but the values of discharges returned to previous values.

As for the transformer temperature, the maximum operating value stipulated by the manufacturer is up to 70 °C and considered critical from 70 °C to 120 °C; during the analysis period, the maximum value measured at Glicerio Avenue was 52 °C, about 75% of the transformer’s normal operating temperature. For the condominium, the maximum value measured was 50 °C, about 70% of the temperature in the normal operation of the transformer. It is also possible to note that the transformer temperature has a strong correlation with the environmental temperature since this type of transformer is mounted on the ground and is exposed to variations in the environmental temperature. 

The maximum current on the low-voltage side of the transformer on Glicerio Avenue is 787.4 A, as stipulated by the manufacturer. The maximum current measured was 280 A in phase A, 180 A in phase B, and 212 A in phase C; these values are below the nominal value. The condominium transformer is similar to the one on Glicerio Avenue. The maximum current measured was 135 A in phase A, 174 A in phase B, and 165 A in phase C.

## 5. Conclusions

In this paper, we provide the results of an optical multi-parameter monitoring solution, describing two medium-voltage underground distribution networks evaluated for 6 months. 

As described in Section 3, the FBG sensors worked accordingly. The partial discharge sensors worked in a non-ideal condition due to the low load current in the medium-voltage conductors, detecting only high levels of partial discharges. An improvement in the harvesting circuit must be achieved for the sensor to work in low current conditions. 

The monitoring system worked accordingly and only stopped working during maintenance or periods lacking electrical energy when the interrogator was turned off.

Regarding the results, anomalies were not observed in the underground distribution networks but were observed in some events of abnormal liquid levels, higher partial discharges, and elevated temperatures in the conductor ducts that exceeded 20 °C in some periods. According to Brazilian standards [29], in these conditions, the maximum supported current specified by the conductor must be corrected to a lower current. 

The monitoring solution is robust enough to allow the detection of any anomaly over a long period of evaluation; in this way, it can anticipate possible failures, reduce inspection and maintenance costs, and optimize underground distribution network assets. The following possible failure can be detected:Real-time conductor loading control;Real-time transformer loading control;Undue intrusion detection;Knowledge of submerged points (manholes);Knowledge of partial discharge evolution.

In addition, the monitoring solution gives a history of the demand for transformers and secondary circuits, allowing the cost optimization of new projects (CAPEX) for underground networks.

## Figures and Tables

**Figure 1 sensors-23-05066-f001:**
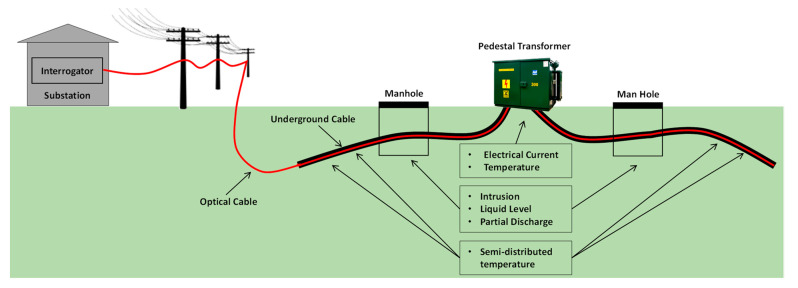
The proposed monitoring system.

**Figure 2 sensors-23-05066-f002:**
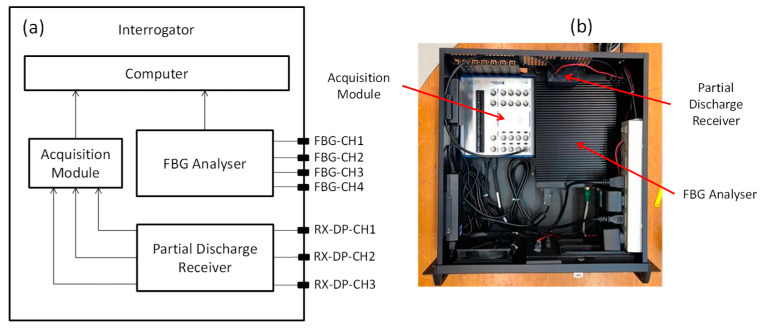
(**a**) Interrogator block diagram and (**b**) interrogator modules.

**Figure 3 sensors-23-05066-f003:**
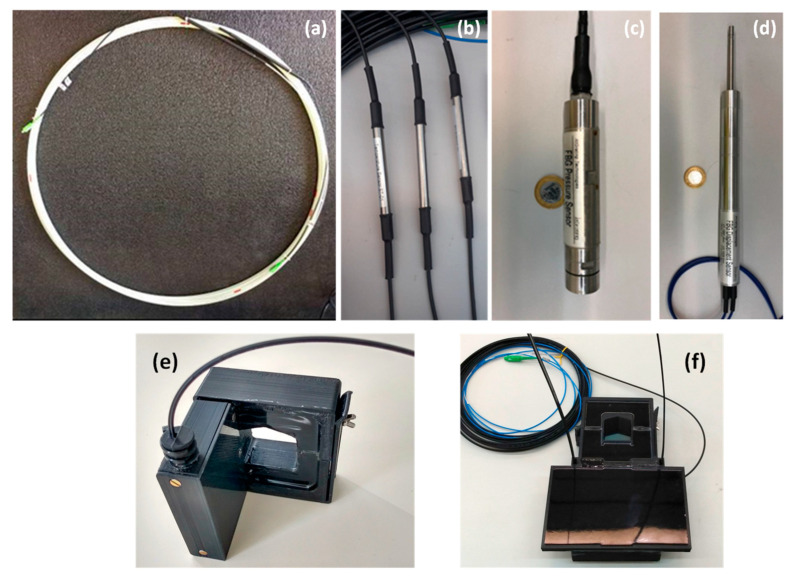
(**a**) Semi-distributed temperature sensor, (**b**) temperature sensor, (**c**) liquid level sensor, (**d**) intrusion sensor, (**e**) current sensor, and (**f**) partial discharge sensor.

**Figure 4 sensors-23-05066-f004:**
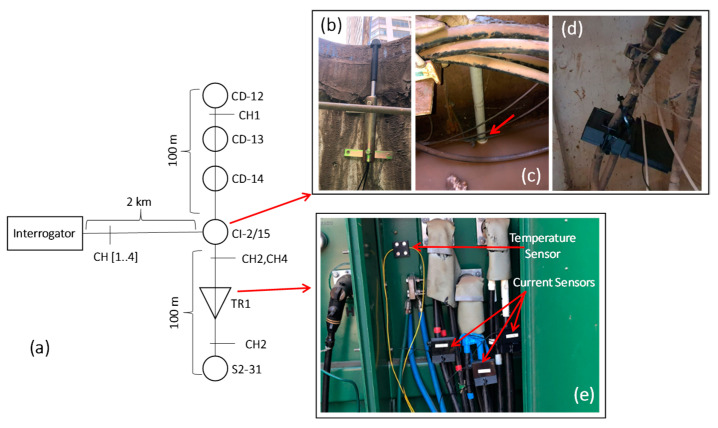
Downtown underground network monitoring, (**a**) single-line diagram, (**b**) intrusion sensor, (**c**) liquid level sensor, (**d**) PD-LD sensor, and (**e**) temperature and current sensors.

**Figure 5 sensors-23-05066-f005:**
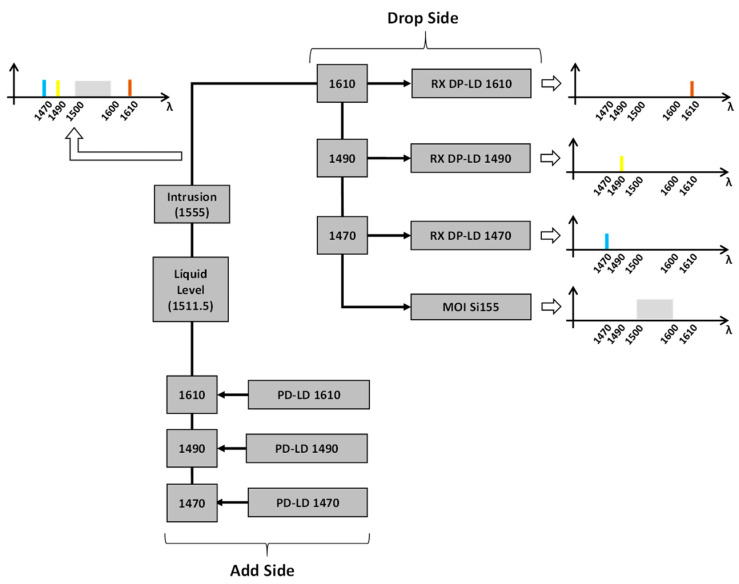
Channel 3 optical configuration with add-drops multiplexing (add-side) and demultiplexing (drop-side).

**Figure 6 sensors-23-05066-f006:**
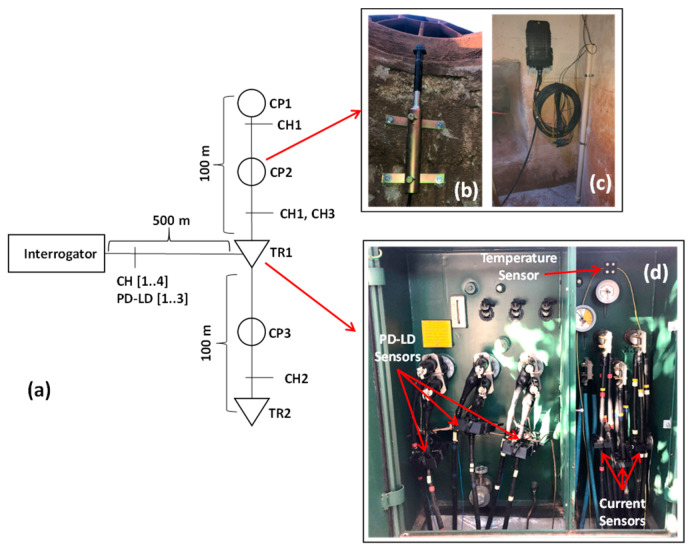
Condominium underground network monitoring, (**a**) single-line diagram, (**b**) intrusion sensor, (**c**) liquid level sensor, (**d**) PD-LD sensor, temperature, and current sensors.

**Figure 7 sensors-23-05066-f007:**
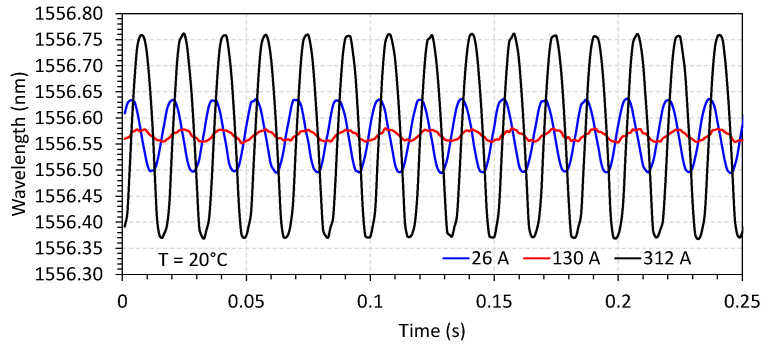
Optical response of FBG current sensor S1 for a fixed temperature of 20 °C and varying the current. In the plot, the current was set to 26, 130, and 312 A.

**Figure 8 sensors-23-05066-f008:**
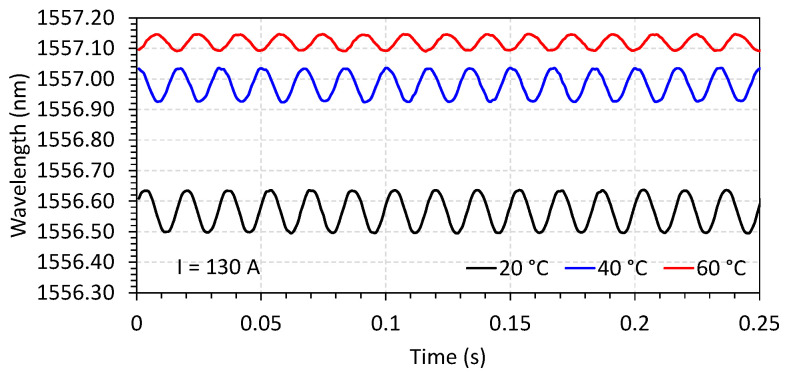
Optical response of FBG current sensor S1 for a fixed current of 130 A, and varying temperatures. In the plot, the temperature was set to 20, 40, and 60 °C.

**Figure 9 sensors-23-05066-f009:**
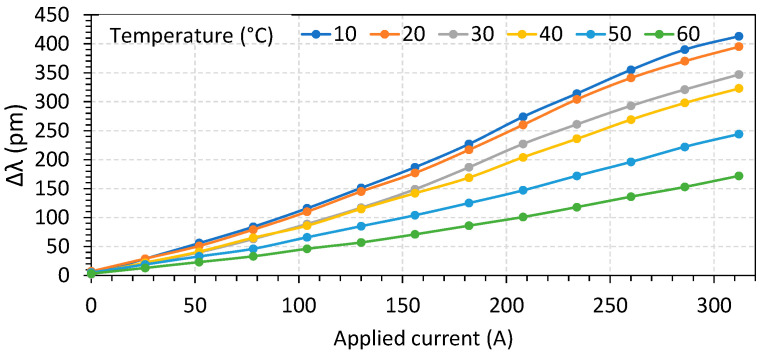
Parameter Δλ as a function of temperature and applied current.

**Figure 10 sensors-23-05066-f010:**
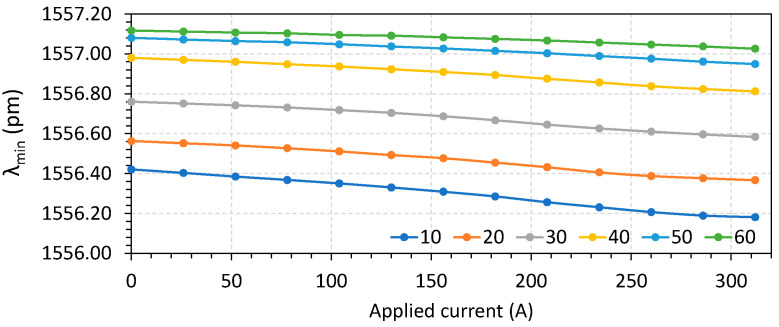
Parameter λmin as a function of temperature and applied current.

**Figure 11 sensors-23-05066-f011:**
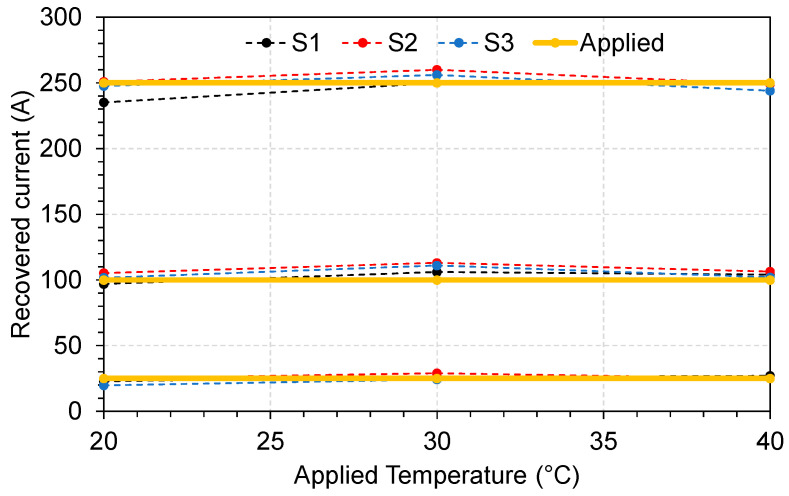
Recovered current for different applied temperatures.

**Figure 12 sensors-23-05066-f012:**
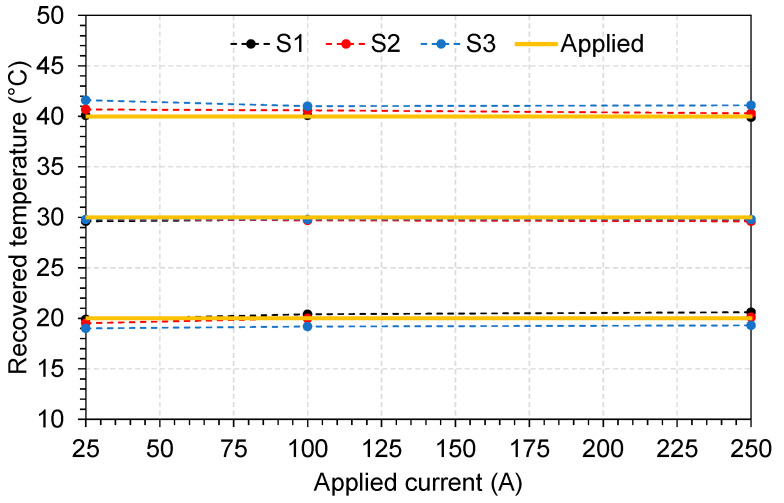
Recovered temperature for different applied currents.

**Figure 13 sensors-23-05066-f013:**
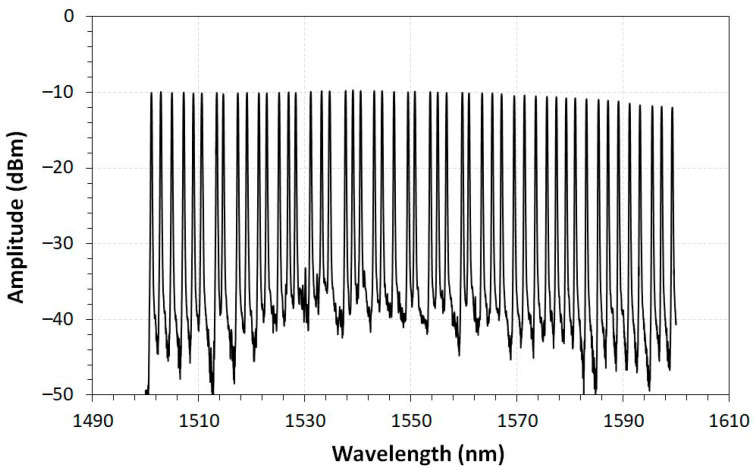
FBG array spectrum with 50 points of temperature measurement.

**Figure 14 sensors-23-05066-f014:**
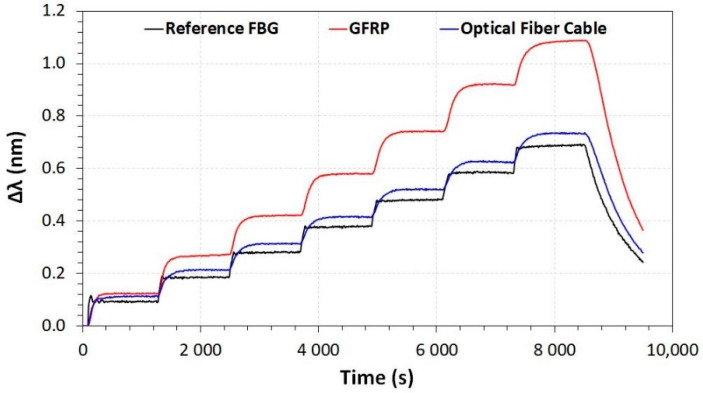
Wavelength shifting as a function of elapsed time for different FBG array cables.

**Figure 15 sensors-23-05066-f015:**
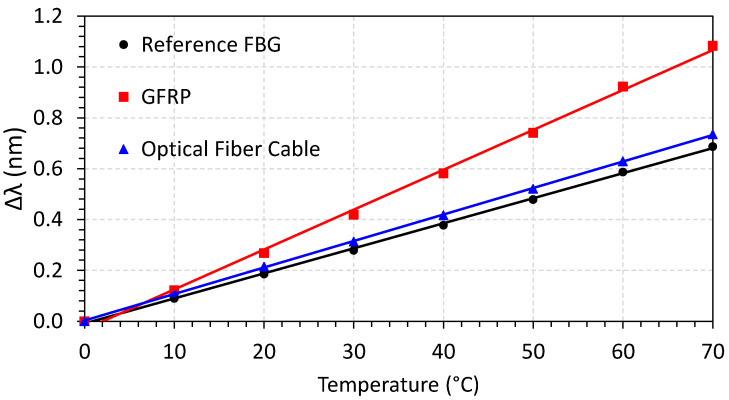
Wavelength shifting as a function of temperature for different FBG array cables.

**Figure 16 sensors-23-05066-f016:**
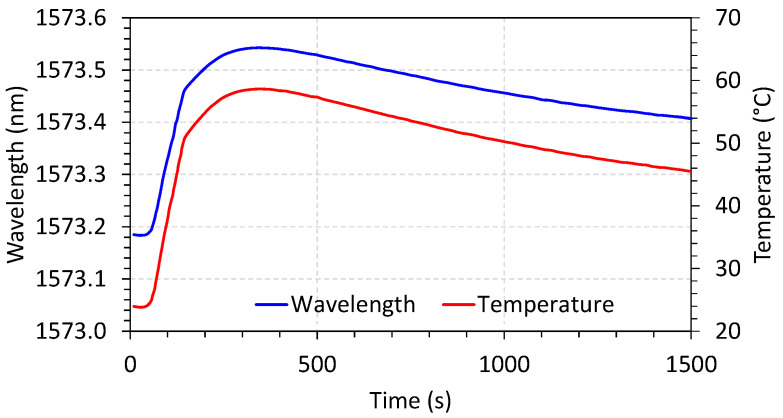
Wavelength displacement and correspondent temperature as a function of time.

**Figure 17 sensors-23-05066-f017:**
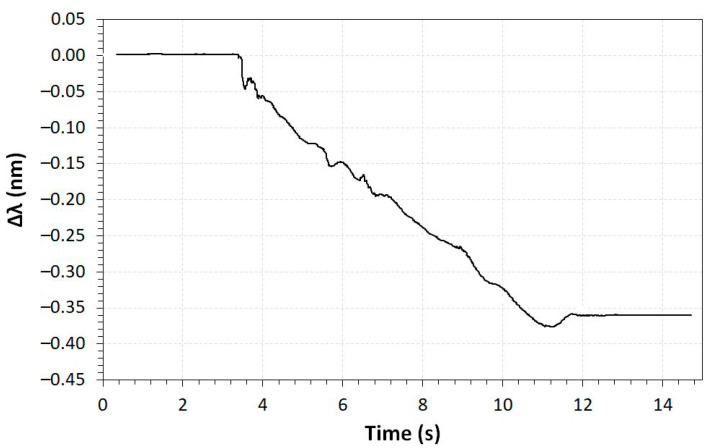
Wavelength shifting as a function of time for the liquid level sensor.

**Figure 18 sensors-23-05066-f018:**
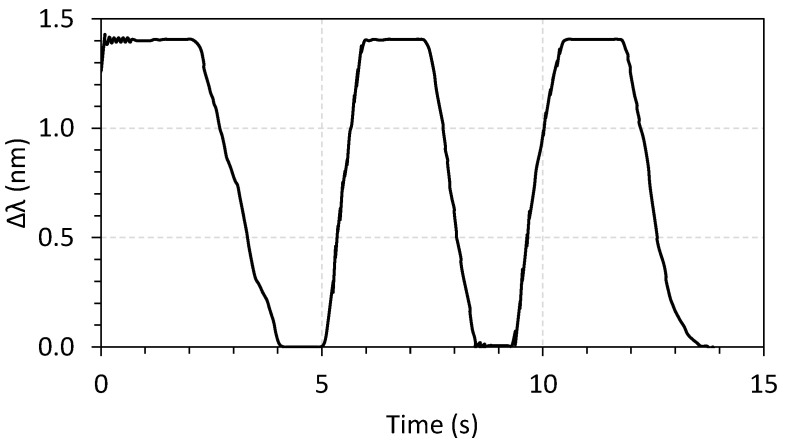
Wavelength shifting as a function of time for the intrusion sensor.

**Figure 19 sensors-23-05066-f019:**
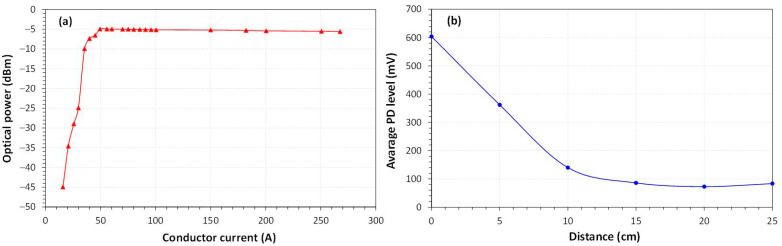
PD-LD sensor behavior: (**a**) optical power as a function of conductor current; (**b**) average PD level as a function of distance.

**Figure 20 sensors-23-05066-f020:**
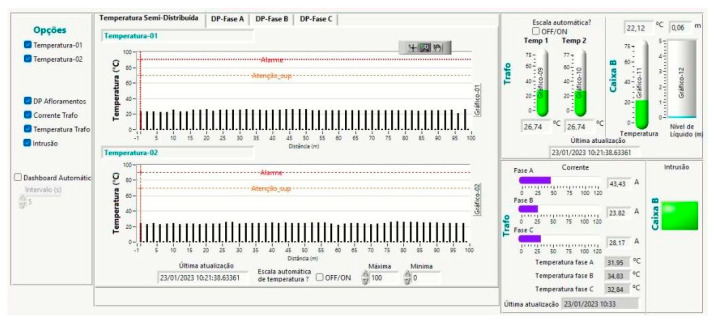
Data acquisition and visualization tool developed in LabVIEW—Semi-distributed temperature.

**Figure 21 sensors-23-05066-f021:**
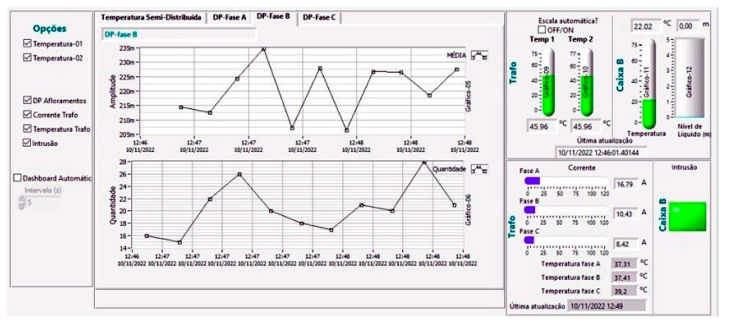
Data acquisition and visualization tool developed in LabVIEW—Partial Discharge.

**Figure 22 sensors-23-05066-f022:**
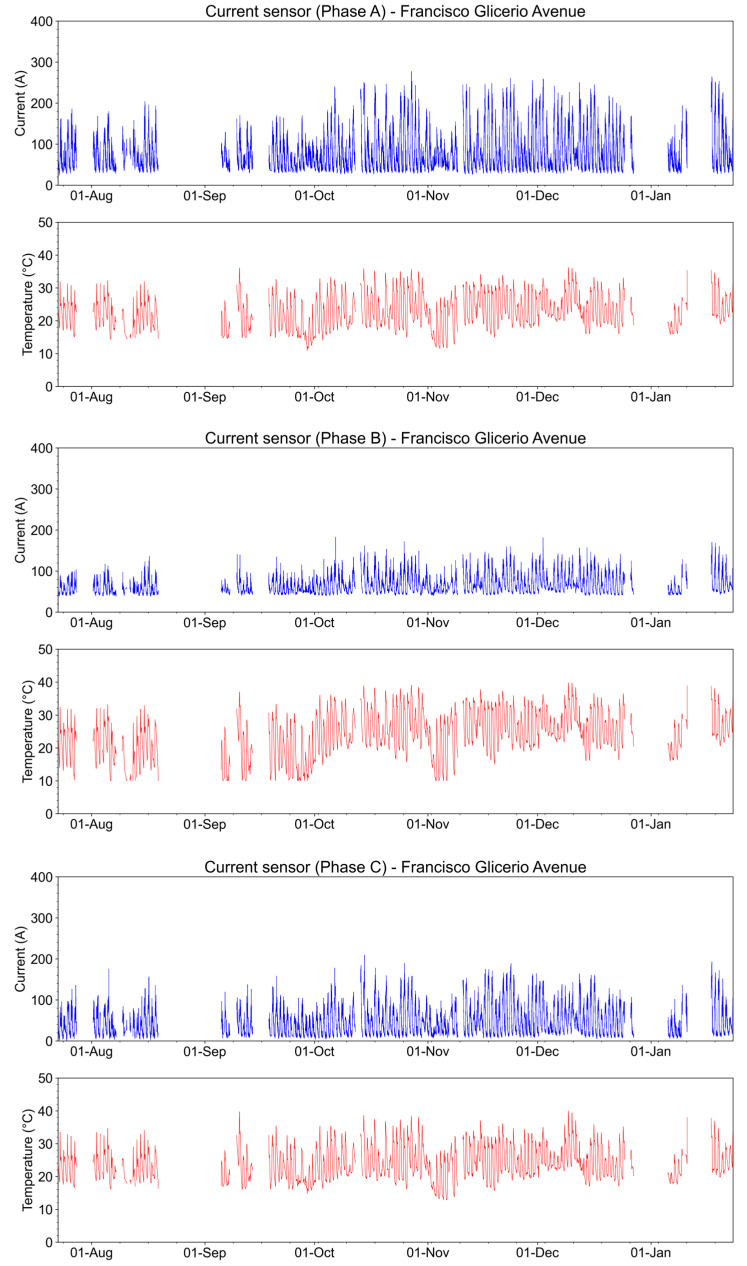
Current and temperature in the secondary winding of the transformer from the end of July 2022 to January 2023.

**Figure 23 sensors-23-05066-f023:**
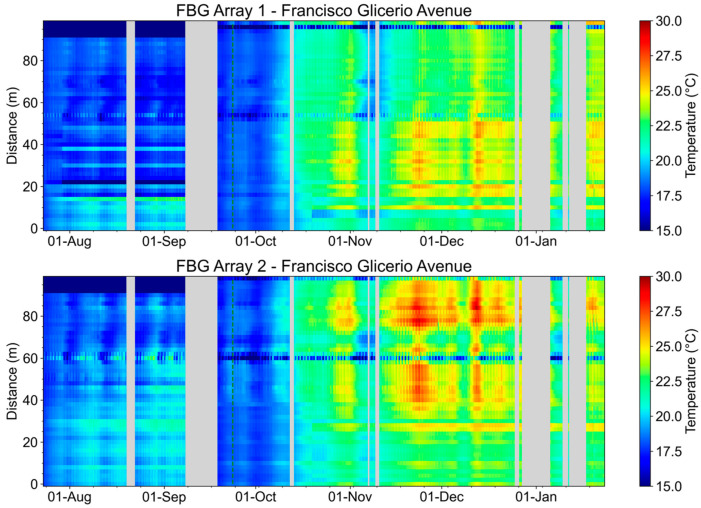
FBG Array 1 and FBG Array 2. Measurements from August 2022 to January 2023. The beginning of the spring in the southern hemisphere is also indicated by the dashed green line.

**Figure 24 sensors-23-05066-f024:**
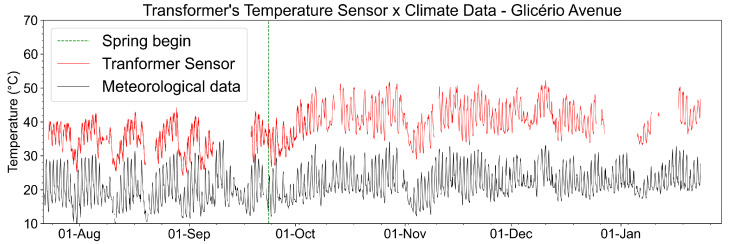
Recovered temperature on the surface of the TPU300 transformer #6 compared with meteorological data of the period. The beginning of spring in the southern hemisphere is also indicated by the dashed green line.

**Figure 25 sensors-23-05066-f025:**
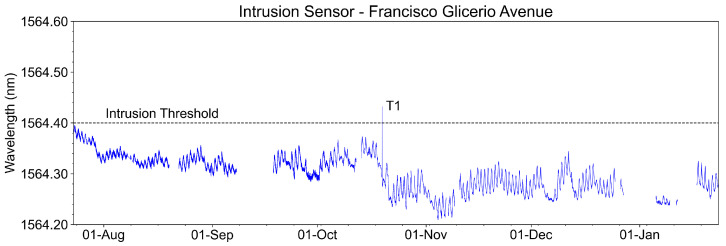
Recovered wavelength of the intrusion sensor at Francisco Glicerio Avenue with one intrusion test event on October 19th, event T1.

**Figure 26 sensors-23-05066-f026:**
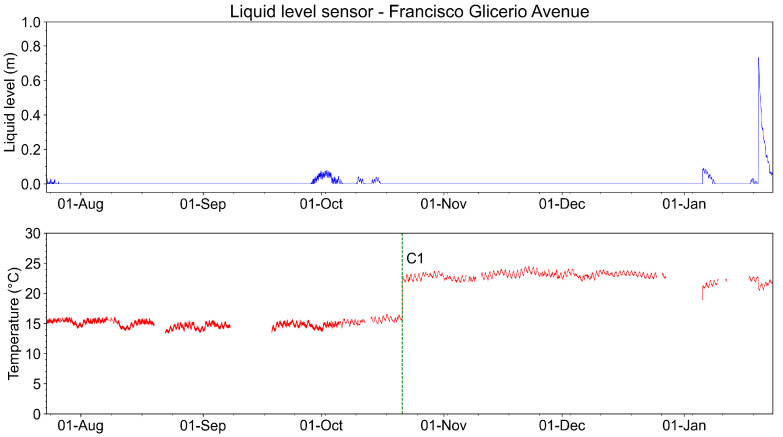
Recovered liquid level and temperature measurements from August to January 2023.

**Figure 27 sensors-23-05066-f027:**
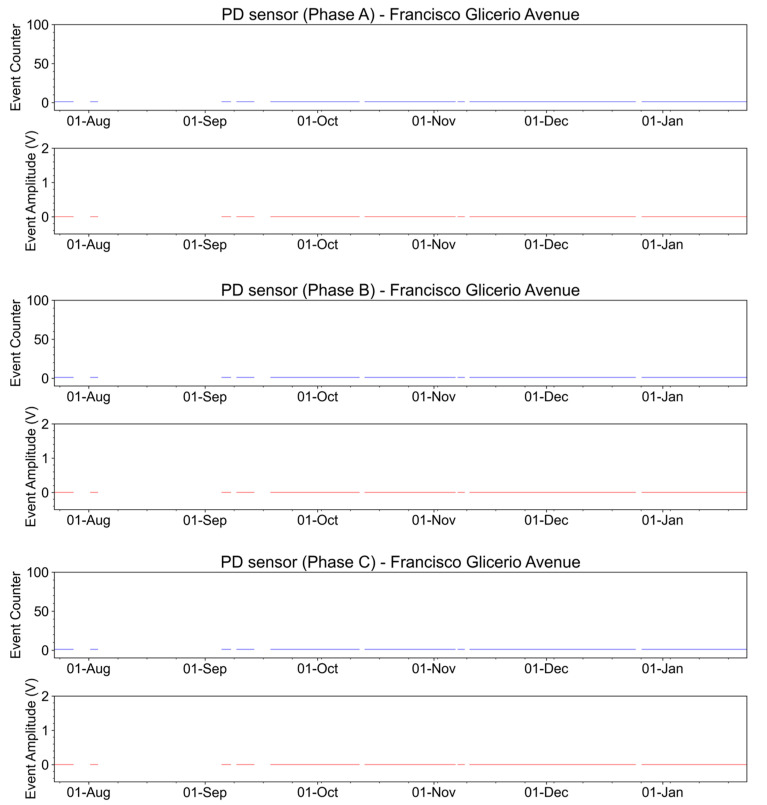
Event counter and amplitude for partial discharge measurements on Francisco Glicerio Avenue from the end of July 2022 to January 2023.

**Figure 28 sensors-23-05066-f028:**
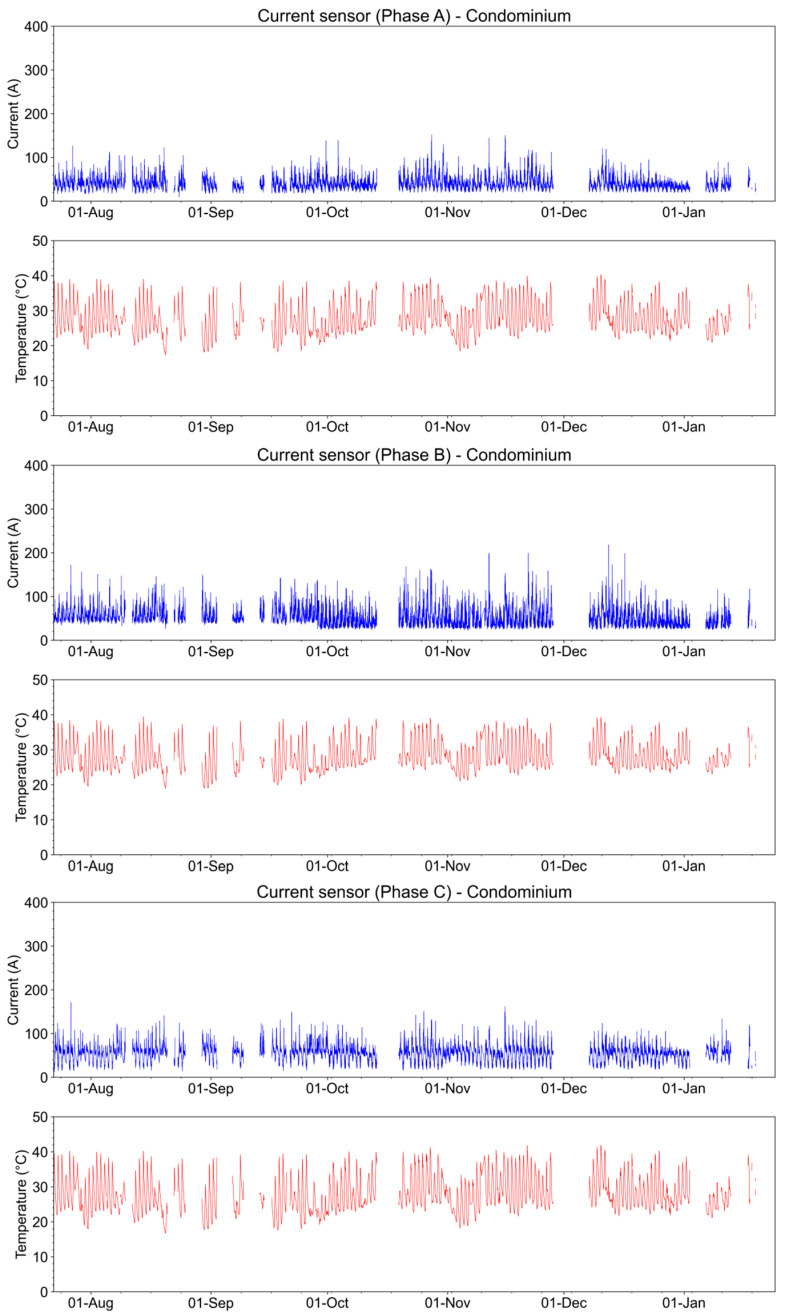
Current and temperature in the secondary of the transformer from the end of July 2022 to January 2023.

**Figure 29 sensors-23-05066-f029:**
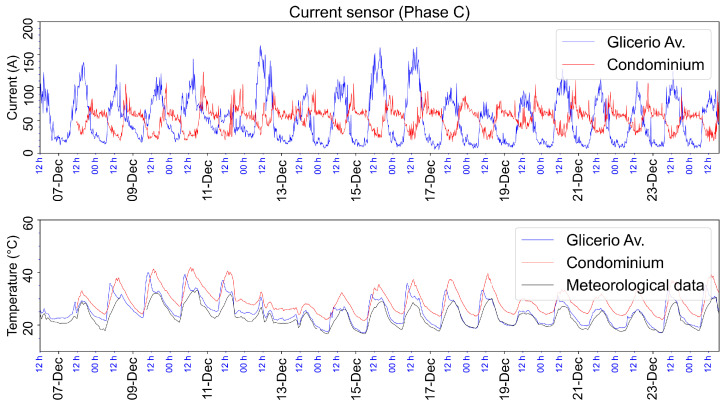
Comparison of data for the condominium and Glicerio Avenue of the recorded current and temperature in the secondary of the transformer (Phase C) from the end of July 2022 to January 2023. Also shown are the meteorological data.

**Figure 30 sensors-23-05066-f030:**
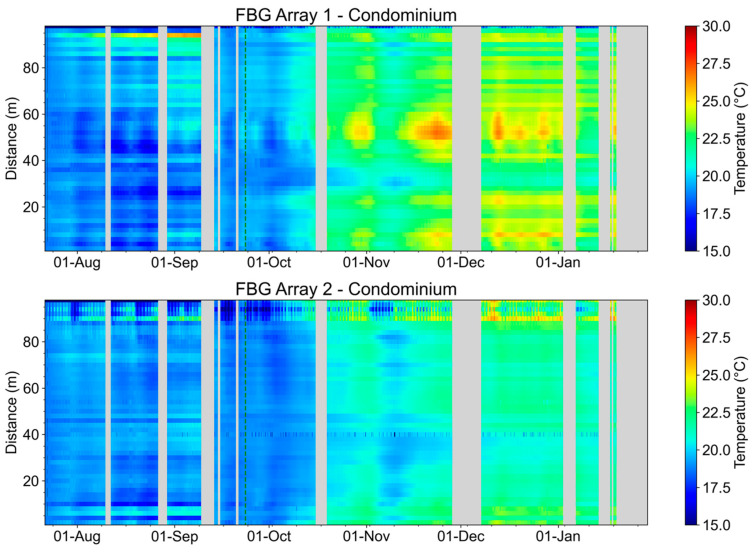
FBG Array 1 and FBG Array 2 measurements from August 2022 to January 2023. The dotted green line indicates the beginning of spring.

**Figure 31 sensors-23-05066-f031:**
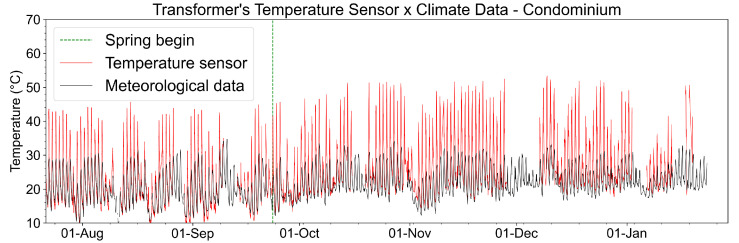
Recovered temperature on the surface of Transformer 1 compared with meteorological data of the period. The beginning of spring in the southern hemisphere is also indicated.

**Figure 32 sensors-23-05066-f032:**
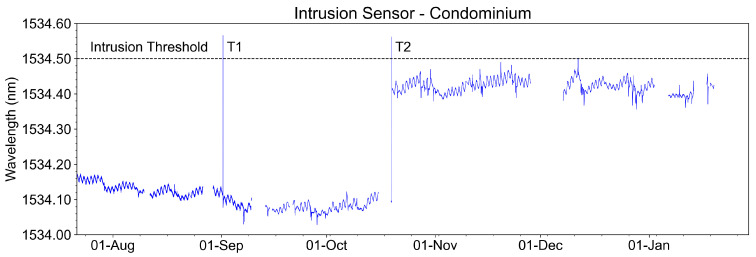
Recovered wavelength of the intrusion sensor at the condominium with two intrusion test events in earlier September, event T1, and late October, event T2.

**Figure 33 sensors-23-05066-f033:**
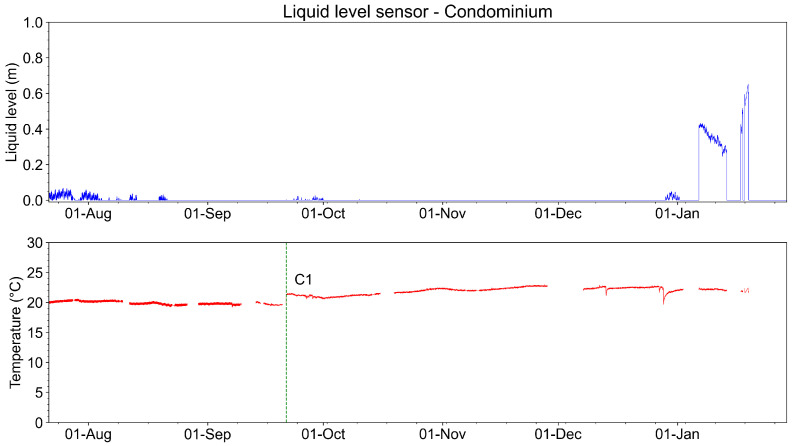
Recorded liquid level and temperature measurements from August 2022 to January 2023.

**Figure 34 sensors-23-05066-f034:**
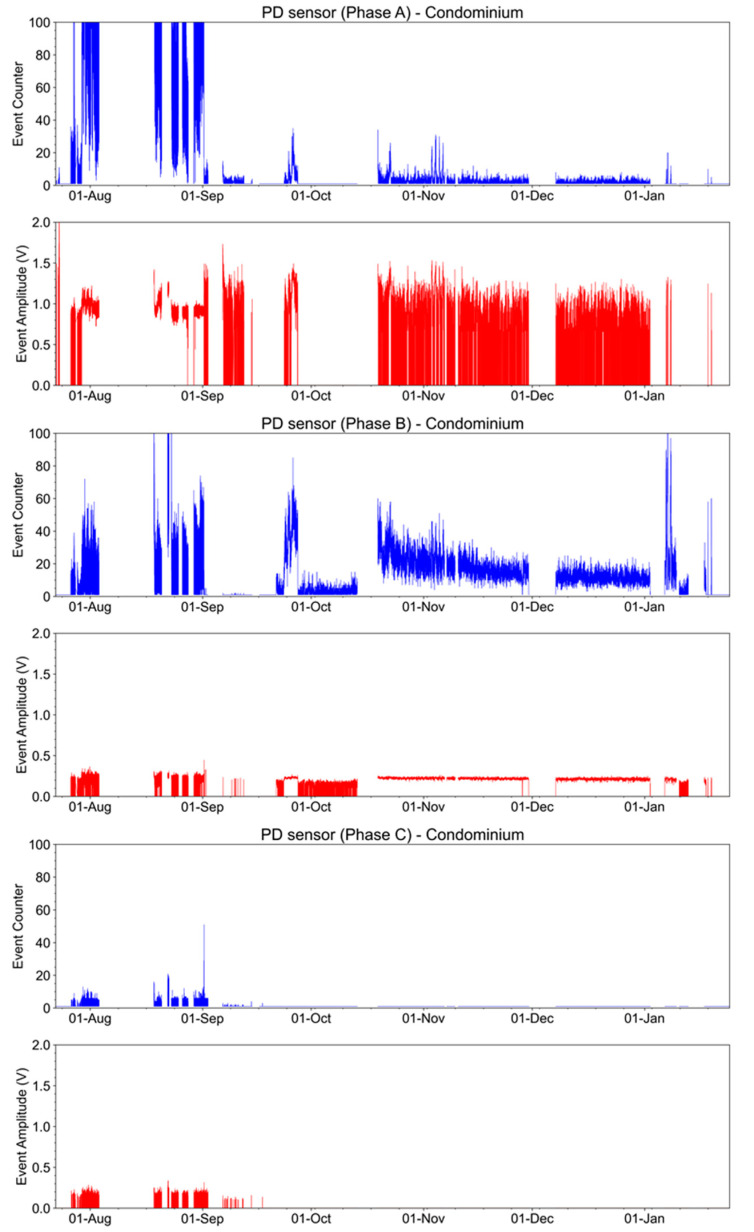
Event counter and amplitude for partial discharge measurements at the condominium from the end of July 2022 to January 2023.

**Table 1 sensors-23-05066-t001:** Interrogator channel, sensors, installation place, and wavelength reference of downtown underground distribution monitoring.

Interrogator Channel	Sensor	Installation Place	Reference Wavelength (nm)
1	Temperature FBG Array (50 FBG, 2 m space)	CI-2/15 → CD-12	1500.5–1598.5
2	Temperature FBG Array (50 FBG, 2 m space)	CI-2/15 → S2-31	1500.5–1598.5
3	Liquid Level	CI-2/15	1511.5
3	Intrusion	CI-2/15	1555
3	PD-LD Phase 1	CI-2/15	1470
3	PD-LD Phase 2	CI-2/15	1490
3	PD-LD Phase 3	CI-2/15	1610
4	Temperature	TR1	1524
4	Current Phase 1	TR1	1556
4	Current Phase 2	TR1	1558
4	Current Phase 3	TR1	1568

**Table 2 sensors-23-05066-t002:** Interrogator channel, sensors, installation place, and wavelength reference of condominium underground distribution monitoring.

Interrogator Channel	Sensor	Installation Place	Reference Wavelength (nm)
1	Temperature FBG Array (50 FBG, 2 m space)	TR1 → CP1	1500.5–1598.5
2	Temperature FBG Array (50 FBG, 2 m space)	TR1 → TR2	1500.5–1598.5
3	Liquid Level	CP2	1539.5
3	Intrusion	CP2	1530
4	Temperature	TR1	1526.5
4	Current Phase 1	TR1	1560
4	Current Phase 2	TR1	1562
4	Current Phase 3	TR1	1571
RX PD-LD 1	PD-LD Phase 1	TR1	1470
RX PD-LD 2	PD-LD Phase 2	TR1	1490
RX PD-LD 3	PD-LD Phase 3	TR1	1610

**Table 3 sensors-23-05066-t003:** FBG arrays monitoring Francisco Glicerio Avenue.

	FBG Array 1	FBG Array 2
Interrogator channel	Channel 1	Channel 2
Monitoring path	CI-2/15–CD-14–CD-13–CD-12 *	Trafo 1–CP3–Trafo 2 **
FBG spatial distance	2 m	2 m
Spectral window	1500–1600 nm	1500–1600 nm

* FBG1 to FBG10 were left outside the electrical cable duct inside the CI-2/15, FBG1 to FBG7 are outside the micro duct, FBG8 to FBG10 are inside the micro duct. ** FBG1 to FBG18 were left outside the electrical cable duct inside the CI-2/15, FBG1 to FBG15 are outside the micro duct, and FBG16 to FBG18 are inside the micro duct.

**Table 4 sensors-23-05066-t004:** Semi-distributed temperature monitoring of the condominium.

	FBG Array 1	FBG Array 2
Interrogator channel	Channel 1	Channel 2
Monitoring path	CP1–CP2–Trafo 1 *	Trafo 1–CP3–Trafo 2 **
FBG spatial distance	2 m	2 m
Spectral window	1500–1600 nm	1500–1600 nm
FBG inside cable duct	50	

* The section between Trafo 1 and CP2 is in a vacant duct (without electrical cables), and part of the FBGs are inside CP2 (FBG28 to FBG33). ** FBG1 is outside the duct, under Transformer 1, and FBG46 to FBG50 are outside the duct, under Transformer 2.

## Data Availability

Data are contained within the article.

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
