# Peer review of "Multi-Parameter Optical Monitoring Solution Applied to Underground Medium-Voltage Electric Power Distribution Networks"

_sensors, 2023, doi:10.3390/s23115066_

Round 1
Reviewer 1 Report
This manuscript reports the multi-parameter optical fiber constructed monitoring solution with the application to the underground medium voltage electric power distribution networks. First, authors describe the laboratory characterization of these sensors based on Fiber Bragg Grating for detailed parameters such as electric current, temperature, liquid level and intrusion (contact sensors) and then they move to the transposition in real condition monitoring in some avenues in the city of Campinas, Brazil. I find this study well-designed and conducted in systematic way, adding the real-time monitoring the very applicative component of this project. Nevertheless, the part of Results is a bit long and rather redundant in the sub-paragraph 3.2.2.1 giving the enumeration of applied parameters rather than the results of this study. I recommend Authors revisiting of this part to make it cleared for readers. Moreover, I have got some remarks which help reader the understanding of these results:
1. Could you explicate the origin and characteristic of FBG sensors used for this study?
2. Some figures, ex. Fig. 22, Fig. 27 and Fig. 34 could be merged to make this manuscript clearer.
3. Which kind of calibration instrument was used for the liquid level sensor (line 495)?
4. Why such a high temperature variation between the Condominium and the meteorological data presented in the Fig. 29.
5. Some data resuming the properties and action of FBG Arrays (lines 557 to 576) could be gather in scheme or table to facilitate reading.
The quality of English is very good, the manuscript is written in a fluid and dynamic language. There are some little misspelling or repetitions in the Introduction part, ex. in the chapter 1, paragraph 5 (lines 55-69) give an impression of redundant listing of references.
In the line 782 :
Rosolem, Joao B. wt al, - it should be replaced by et al.
Author Response
Comments and Suggestions for Authors
This manuscript reports the multi-parameter optical fiber constructed monitoring solution with the application to the underground medium voltage electric power distribution networks. First, authors describe the laboratory characterization of these sensors based on Fiber Bragg Grating for detailed parameters such as electric current, temperature, liquid level and intrusion (contact sensors) and then they move to the transposition in real condition monitoring in some avenue sin the city of Campinas, Brazil. I find this study well-designed and conducted in systematic way, adding the real-time monitoring the very applicative component of this project. Nevertheless, the part of Results is a bit long and rather redundant in the sub-paragraph 3.2.2.1 giving the enumeration of applied parameters rather than the results of this study. I recommend Authors revisiting of this part to make it cleared for readers. Moreover, I have got some remarks which help reader the understanding of these results:
1. Could you explicate the origin and characteristic of FBG sensors used for this study?
R. All FBG sensors were supplied by Atgrating, they also supplied the FBG sensors datasheet with all FBG sensors characteristics. We also have previous experience with extensometers and liquid level based on FBGs, this experience is reported in [19]. We improve the paragraph in section 2.3 with this new information. The new text below was inserted in the manuscript (in red) [line 157-160]:
The FBG array, temperature, liquid level, and intrusion sensors are commercial parts, current sensor it’s a homemade development, but uses a raw FBG. All FBG sensors and raw FBGs were supplied by Atgrating, they also supplied the FBG sensors datasheet with all FBG sensors characteristics.
2. Some figures, ex. Fig. 22, Fig. 27 and Fig. 34 could be merged to make this manuscript clearer.
R. We thank your suggestion, but to demonstrate the behavior of the three phases we believe that Fig. 22, Fig. 27, and Fig. 34 are clearer if shown the three phases were separated. We will maintain Fig. 22, Fig. 27, and Fig. 34 separated.
3. Which kind of calibration instrument was used for the liquid level sensor (line 495)?
R. There are two sensors inside the liquid level sensor, the liquid level and temperature sensors. The temperature sensor is used mainly to correct the liquid level measurement, but we also use the temperature sensor to measure the temperature inside the manhole. The angular sensor coefficients are supplied by the sensor datasheet and the calibration is necessary because of the linear coefficient (offset) that changes according to the distance between the sensor and the interrogator.
Liquid level calibration was performed using a measuring tape, where we measured the distance between the manhole cover and the water level inside and subtract this value from the total depth of the manhole, and the temperature sensor calibration was made using a thermopile close to the liquid level sensor. As mentioned before the calibration was necessary due to the offset deviation caused by the distance between the interrogator and the sensor. We also improve the paragraph with this new information. The new text below was inserted in the manuscript (in red) [line 505-509]:
Figure 26 shows the measurements of the liquid level sensor (in meters) and its temperature from August 2022 to January 2023. We highlight in the graph a calibration performed in the temperature measurement (C1). The temperature adjustment was made using a calibration instrument (thermopile). The difference found was +6.6 °C. The compensation was made by adding +6.6°C to the system measurement.
4. Why such a high temperature variation between the Condominium and the meteorological data presented in the Fig. 29.
R. Both the condominium and venue current sensors are placed inside a pad-mounted transformer to measure the current and temperature from the low-voltage side.
Two factors explain the temperature behavior, sun exposition, and load characteristic of the transformers. If on one hand, the transformer at Avenue is placed below some trees and did not receive sun radiation during the day, furthermore, the load characteristic of the transformer is high during the day and low during the night due to this place being a commercial center and there is more electrical consumption during the day. On the other hand, a transformer at the condominium receives direct sun radiation, and during the night there is an increase in the load due to the higher consumption at night when people return from work activities to their homes and use showers and air conditioners. The sun radiation is responsible for the temperature offset during the day and the load of the transformer is responsible for the temperature offset during the night.
In addition, we also improve the paragraph before Fig.29 with this new information. The new text below was inserted in the manuscript (in red) [line 563-572]:
Two factors explain the temperature behavior, sun exposition, and load characteristic of the transformers. If on one hand, the transformer at Avenue is placed below some trees and did not receive sun radiation during the day, furthermore, the load characteristic of the transformer is high during the day and low during the night due to this place being a commercial center and there is more electrical consumption during the day. On the other hand, a transformer at the condominium receives direct sun radiation, and during the night there is an increase in the load due to The higher consumption at night when people return from work activities to their homes and use high-consumption appliances. The sun radiation is responsible for the temperature offset during the day and the load of the transformer is responsible for the temperature offset during the night.
5. Some data resuming the properties and action of FBG Arrays (lines 557 to 576) could be gather in scheme or table to facilitate reading.
R. Thank you for the comment, we added Table 3 and Table 4 to the manuscript resuming the properties and action of FBG Arrays for Avenue and Condominium respectively. The new tables are highlighted in red on the manuscript [line 446-453; 586-592].
6. Comments on the Quality of English Language
The quality of English is very good, the manuscript is written in a fluid and dynamic language. There are some little misspelling or repetitions in the Introduction part, ex. in the chapter 1, paragraph 5 (lines 55-69) give an impression of redundant listing of references.
In the line 782: Rosolem, Joao B. wt al, - it should be replaced by et al.
R. Ok, we correct it.

Reviewer 2 Report
This work presents a multi-parameter optical fiber monitoring solution applied to an underground power distribution network. The following problems still exist, and the author is suggested to revise them carefully.
1. In the instruction, what are the main contributions of this paper? It is recommended that the author extract 2-3 contributions of this paper.
2. What are the advantages compared to previous methods? Such as detection time and accuracy.
3. “In this way, temperature sensing along the conductive cable can indicate punctual faults in the cable, or network elements and is useful for verifying the ampacity of the circuit being monitored.” What are the fault features when a fault occurs?
4. It is suggested to add the following relevant research Or related application fields,and it is also recommended that the author add some references of MDPI.
[1] Chen, S.; Wang, J.; Zhang, C.; Li, M.; Li, N.; Wu, H.; Liu, Y.; Peng, W.; Song, Y. Marine Structural Health Monitoring with Optical Fiber Sensors: A Review. Sensors 2023, 23, 1877. https://doi.org/10.3390/s23041877
[2] Kou, L.; Li, Y.; Zhang, F.; et al. Review on Monitoring, Operation and Maintenance of Smart Offshore Wind Farms. Sensors 2022, 22, 2822. https://doi.org/10.3390/s22082822
[3] Yan, XS; Hu, CY and Sheng, VS. Data-driven pollution source location algorithm in water quality monitoring sensor networks. International Journal of Bio-Inspired Computation,2020, 15 (3),171-180
[4] Minutolo, V.; Cerri, E.; Coscetta, A.; Damiano, E.; De Cristofaro, M.; Di Gennaro, L.; Esposito, L.; Ferla, P.; Mirabile, M.; Olivares, L.; Zona, R. NSHT: New Smart Hybrid Transducer for Structural and Geotechnical Applications. Appl. Sci. 2020, 10, 4498. https://doi.org/10.3390/app10134498
Author Response
Comments and Suggestions for Authors
This work presents a multi-parameter optical fiber monitoring solution applied to an underground power distribution network. The following problems still exist, and the author is suggested to revise them carefully.
1. In the introduction, what are the main contributions of this paper? It is recommended that the author extract 2-3 contributions of this paper.
R. Thanks for your comment, we improve the paragraph regarding the paper contribution in the introduction. The new text below was inserted in the manuscript (in blue) [lines 115-122]:
The main contribution of this paper, to the best of our knowledge, is the first demonstration of an optical multi-parameter system applied to a medium voltage distribution network using FBG sensors with laboratory and field results. We also highlight the method that permits a current sensor with temperature compensation using terfenol-D with a single FBG, and the optical network topology deployed that allows partial discharge sensor based on energy harvesting and laser as light emission share the same optical fiber of sensors based on light reflection (FBG).
2. What are the advantages compared to previous methods? Such as detection time and accuracy.
R. There are several advantages intrinsic to optical fibers, such as high sensitivity, immunity to electromagnetic interference, wide bandwidth, robustness in harsh environments, ease of installation, long lifespan, and for FBG sensors the network-multiplexing capability of different sensors in a single-fiber is also an advantage of optical sensors. In addition, electrical power from alternated current (AC), batteries, and solar panels are not necessary to supply the sensors.
In terms of detection time, the optical fiber sensor's advantage rises from the network's reliability as the sensor information is inside the optical fiber and the distance or line-of-sight between sensor and interrogator is not a problem as it is for wireless sensors for example. Network availability is also an advantage for optical sensors in terms of detection time.
Concerning accuracy, FBG sensors can detect even small changes in physical parameters, and in addition, the information measured by the sensor is in the wavelength variation (frequency) and not in the amplitude, which maintains the fidelity of the measurement without losses along the optical fiber due to signal attenuation. Accuracy is also linked to the ability to detect small variations in the wavelength reflected by the FBG, in this case, we have a resolution of 0.1°C for the temperature sensor.
In addition, we also improve the paragraph in the introduction. The new text below was inserted in the manuscript (in blue) [lines 85-98]:
There are several advantages intrinsic to optical fibers, such as high sensitivity, immunity to electromagnetic interference, wide bandwidth, resistance to harsh environments, ease of installation, long lifespan, and for FBG sensors the network-multiplexing capability of different sensors in a single-fiber is also an advantage of optical sensors. In addition, electrical power from alternated current (AC), batteries, and solar panels are not necessary. In terms of detection time, the optical fiber sensor's advantage rises from the network's reliability as the sensor information is inside the optical fiber and the distance or line-of-sight between sensor and interrogator is not a problem as it is for wireless sensors for example. Network availability is also an advantage for optical sensors in terms of detection time. Concerning accuracy, FBG sensors can detect even small changes in physical parameters, and in addition, the information measured by the sensor is in the wavelength variation (frequency) and not in the amplitude, which maintains the fidelity of the measurement without losses along the optical fiber due to signal attenuation.
3. “In this way, temperature sensing along the conductive cable can indicate punctual faults in the cable, or network elements and is useful for verifying the ampacity of the circuit being monitored.” What are the fault features when a fault occurs?
R. A punctual fault in the cable could affect the conductivity of the conductor material (cooper or aluminum) at this point. The conductive change will increase the temperature and can degrade the cable insulation causing current leakage and even a short circuit between the conductor and armoring of the medium voltage cable. Monitoring the temperature along the cable can prevent current leakage and even a short circuit, avoiding electrical network unavailability, and allows optimizing the cable ampacity as the current carrying capacity in the electrical cable is linked to the ambient temperature.
In addition, we also improve the paragraph in the Discussion. The new text below was inserted in the manuscript (in blue) [line 679-686]:
A punctual fault in the cable could affect the conductivity of the conductor material (Cooper or Aluminum) at this point. The conductive change will increase the temperature and can degrade the cable insulation causing current leakage and even a short circuit between the conductor and armoring of the medium voltage cable. Monitoring the temperature along the cable can prevent current leakage and even a short circuit, avoiding electrical network unavailability, and allows optimizing the cable ampacity as the current carrying capacity in the electrical cable is linked to the ambient temperature.
4. It is suggested to add the following relevant research Or related application fields and it is also recommended that the author add some references of MDPI.
[1] Chen, S.; Wang, J.; Zhang, C.; Li, M.; Li, N.; Wu, H.; Liu, Y.; Peng, W.; Song, Y. MarineStructural Health Monitoring with Optical Fiber Sensors: A Review. Sensors 2023, 23, 1877.https://doi.org/10.3390/s23041877
[2] Kou, L.; Li, Y.; Zhang, F.; et al. Review on Monitoring, Operation and Maintenance of SmartOffshore Wind Farms. Sensors 2022, 22, 2822. https://doi.org/10.3390/s22082822
[3] Yan, XS; Hu, CY and Sheng, VS. Data-driven pollution source location algorithm in waterquality monitoring sensor networks. International Journal of Bio-Inspired Computation,2020,15 (3),171-180
[4] Minutolo, V.; Cerri, E.; Coscetta, A.; Damiano, E.; De Cristofaro, M.; Di Gennaro, L.;Esposito, L.; Ferla, P.; Mirabile, M.; Olivares, L.; Zona, R. NSHT: New Smart HybridTransducer for Structural and Geotechnical Applications. Appl. Sci. 2020, 10, 4498.https://doi.org/10.3390/app10134498
R. Thank you for the suggestion. We added the reference [1] and others with related application fields and some MDPI references. The new references are as below and it was and citations were inserted in the manuscript (in blue).
- Khan, N. Malik, A. Al-Arainy and S. Alghuwainem, "A review of condition monitoring of underground power cables," 2012 IEEE International Conference on Condition Monitoring and Diagnosis, Bali, Indonesia, 2012, pp. 909-912, doi: 10.1109/CMD.2012.6416300
- Nakamura, S. Morooka and K. Kawasaki, "Conductor temperature monitoring system in underground power transmission XLPE cable joints," in IEEE Transactions on Power Delivery, vol. 7, no. 4, pp. 1688-1697, Oct. 1992, doi: 10.1109/61.156967.
- O. Hill and G. Meltz, "Fiber Bragg grating technology fundamentals and overview," in Journal of Lightwave Technology, vol. 15, no. 8, pp. 1263-1276, Aug. 1997, doi: 10.1109/50.618320
- Chen, S.; Wang, J.; Zhang, C.; Li, M.; Li, N.; Wu, H.; Liu, Y.; Peng, W.; Song, Y. Marine Structural Health Monitoring with Optical Fiber Sensors: A Review. Sensors 2023, 23, 1877.https://doi.org/10.3390/s23041877
- Swain, Akhyurna, Elmouatamid Abdellatif, Ahmed Mousa, and Philip W. T. Pong. 2022. "Sensor Technologies for Transmission and Distribution Systems: A Review of the Latest Developments" Energies 15, no. 19: 7339. https://doi.org/10.3390/en15197339
- Hyunjin Kim, Siwoong Park, Chanil Yeo, Hyun Seo Kang, and Hyoung-Jun Park "Thermal analysis of 22.9-kV crosslinked polyethylene cable joint based on partial discharge using fiber Bragg grating sensors," Optical Engineering 60(3), 034101 (2 March 2021). https://doi.org/10.1117/1.OE.60.3.034101
- Gonçalves, Marceli N., and Marcelo M. Werneck. 2021. "Optical Voltage Transformer Based on FBG-PZT for Power Quality Measurement" Sensors 21, no. 8: 2699. https://doi.org/10.3390/s21082699
- Elsayed, Yasser, and Hossam A. Gabbar. 2022. "FBG Sensing Technology for an Enhanced Microgrid Performance" Energies 15, no. 24: 9273. https://doi.org/10.3390/en15249273
- IEC 61936-1:2002, “Power installations exceeding 1 kV a.c. - Part 1: Common rules,” 2002.
- NF C13-200, "High voltage electrical installations - Additional rules for production sites and industrial, commercial and agricultural installations", 2009.

Round 2
Reviewer 2 Report
Many references in the article are too old, it is recommended to remove them. And it is also recommended that the author add some references of MDPI.
[1] Minutolo, V.; Cerri, E.; Coscetta, A.; Damiano, E.; De Cristofaro, M.; Di Gennaro, L.;Esposito, L.; Ferla, P.; Mirabile, M.; Olivares, L.; Zona, R. NSHT: New Smart HybridTransducer for Structural and Geotechnical Applications. Appl. Sci. 2020, 10, 4498.https://doi.org/10.3390/app10134498
[2] Kou, L.; Li, Y.; Zhang, F.; et al. Review on Monitoring, Operation and Maintenance of SmartOffshore Wind Farms. Sensors 2022, 22, 2822. https://doi.org/10.3390/s22082822
[3] Yan, XS; Hu, CY and Sheng, VS. Data-driven pollution source location algorithm in waterquality monitoring sensor networks. International Journal of Bio-Inspired Computation,2020,15 (3),171-180
Author Response
Reviewer 2, round 2.
Many references in the article are too old, it is recommended to remove them. And it is also recommended that the author add some references of MDPI.
[1] Minutolo, V.; Cerri, E.; Coscetta, A.; Damiano, E.; De Cristofaro, M.; Di Gennaro, L.;Esposito, L.; Ferla, P.; Mirabile, M.; Olivares, L.; Zona, R. NSHT: New Smart HybridTransducer for Structural and Geotechnical Applications. Appl. Sci. 2020, 10, 4498.https://doi.org/10.3390/app10134498
[2] Kou, L.; Li, Y.; Zhang, F.; et al. Review on Monitoring, Operation and Maintenance of SmartOffshore Wind Farms. Sensors 2022, 22, 2822. https://doi.org/10.3390/s22082822
[3] Yan, XS; Hu, CY and Sheng, VS. Data-driven pollution source location algorithm in water quality monitoring sensor networks. International Journal of Bio-Inspired Computation,2020,15 (3),171-180
R. Thank you for your suggestion. These papers recommended do not describe applications for the power electric sector, such as those required for this Special Issue, but they describe applications for geotechnical, pollution monitoring, and wind farms and because of this they were not cited in the revision of the manuscript.
Furthermore, we added new current references and many of them are from MDPI as recommended by Reviewer 2, round 1. We preferred to cite references from the power electric sector related to the current application described in the manuscript. Please see below:
Chen, S.; Wang, J.; Zhang, C.; Li, M.; Li, N.; Wu, H.; Liu, Y.; Peng, W.; Song, Y. Marine Structural Health Monitoring with Optical Fiber Sensors: A Review. Sensors 2023, 23, 1877.https://doi.org/10.3390/s23041877
Swain, Akhyurna, Elmouatamid Abdellatif, Ahmed Mousa, and Philip W. T. Pong. 2022. "Sensor Technologies for Transmission and Distribution Systems: A Review of the Latest Developments" Energies 15, no. 19: 7339. https://doi.org/10.3390/en15197339
Gonçalves, Marceli N., and Marcelo M. Werneck. 2021. "Optical Voltage Transformer Based on FBG-PZT for Power Quality Measurement" Sensors 21, no. 8: 2699. https://doi.org/10.3390/s21082699
Elsayed, Yasser, and Hossam A. Gabbar. 2022. "FBG Sensing Technology for an Enhanced Microgrid Performance" Energies 15, no. 24: 9273. https://doi.org/10.3390/en15249273
